# Current Status of Mosquito Handling, Transporting and Releasing in Frame of the Sterile Insect Technique

**DOI:** 10.3390/insects13060532

**Published:** 2022-06-10

**Authors:** Jiatian Guo, Xiaoying Zheng, Dongjing Zhang, Yu Wu

**Affiliations:** 1Sun Yat-sen University–Michigan State University Joint Center of Vector Control for Tropical Diseases, Key Laboratory of Tropical Disease Control of the Ministry of Education, Sun Yat-sen University, Guangzhou 510080, China; guojt9@mail2.sysu.edu.cn (J.G.); zhengxy@mail.sysu.edu.cn (X.Z.); 2Chinese Atomic Energy Agency Center of Excellence on Nuclear Technology Applications for Insect Control, Sun Yat-sen University, Guangzhou 510080, China; 3International Atomic Energy Agency Collaborating Center, Sun Yat-sen University, Guangzhou 510080, China; 4Guangdong Provincial Engineering Technology Research Center for Diseases-Vectors Control, Sun Yat-sen University, Guangzhou 510080, China

**Keywords:** mosquito, chilling, quality control, packaging

## Abstract

**Simple Summary:**

With the increasing burden of mosquito-borne diseases around the world and the traditional control methods showing drawbacks, the sterile insect technique (SIT) is now a potential new tool in the field of controlling mosquitoes. During the implementation of SIT, several steps, such as handling, transportation and release, are of great importance and stand a chance to be optimized. Here, we provide an overview of the key steps in the whole SIT process, listing the main handling, transporting and releasing methods described in the present studies in order to maximize the success of the SIT. With the relevant technical summary, the cognition of this technology can be more accurate; the explorations and results may evoke more follow-up research, making them more directional.

**Abstract:**

The sterile insect technique (SIT) and its related technologies are considered to be a powerful weapon for fighting against mosquitoes. As an important part of the area-wide integrated pest management (AW-IPM) programs, SIT can help reduce the use of chemical pesticides for mosquito control, and consequently, the occurrence of insecticide resistance. The mosquito SIT involves several important steps, including mass rearing, sex separation, irradiation, packing, transportation, release and monitoring. To enable the application of SIT against mosquitoes to reduce vector populations, the Joint Food and Agriculture Organization of the United Nations (FAO) and the International Atomic Energy Agency (IAEA) Centre (previously called Division) of Nuclear Techniques in Food and Agriculture (hereinafter called Joint FAO/IAEA Centre) and its Insects Pest Control sub-program promoted a coordinated research project (CRP) entitled “Mosquito handling, transport, release and male trapping methods” to enhance the success of SIT. This article summarizes the existing explorations that are critical to the handling and transporting of male mosquitoes, offers an overview of detailed steps in SIT and discusses new emerging methods for mosquito releases, covering most processes of SIT.

## 1. Introduction

Mosquito-transmitted diseases, such as dengue, Zika and chikungunya, have become major global health emergencies in recent years. For example, dengue has now been the world’s most common mosquito-borne disease with more than a 30-fold increase in its occurrence [1]. It is estimated that over half of the world’s population is at risk of dengue [2]. In addition, yellow fever and malaria are re-emerging at the same time [3]. The general strategy of protecting humans and livestock from these pathogens is to find effective medicine or to develop a vaccine; however, the progress in these two strategies is limited. Vector control remains the primary method to prevent mosquito-borne diseases [4]. Classical methods of vector control, which mainly focused on reducing mosquito populations through the application of insecticides or eliminating larval breeding sites, are becoming ineffective and unsustainable. In addition, the overuse of insecticides has reached its limits because of genetic resistance and negative consequence on other species and the environment [5]. Developing novel approaches for mosquito control is urgently required [6].

The sterile insect technique (SIT) is a biologically based method for the management of key insect pests of agricultural and medical/veterinary importance. The SIT is a pest management method that involves releasing sterile male insects in large numbers across a large region to limit reproduction in a field population of the same species [7]. As the emergence of the insects decreases, it is therefore a strategy of “birth control”. Additionally, SIT is “non-GMO (genetically modified organisms)” as it does not involve the release of insects modified through transgenic (genetic engineering) processes. In this type of autocidal control, continuous releases of the sterilized male insects at an adequate ratio of sterile to fertile males will lead to a reduction in pest populations. Effective control using radiation-sterilized male insects is achieved, as it is part of area-wide integrated pest management (AW-IPM) programs. Successful examples are the eradication of the New World screwworm *Cochliomyia hominivorax* in North America [8], the eradication of the tsetse fly *Glossina austeni* in the island of Unguja, Zanzibar [9], and the eradication of the melon fly *Bactrocera cucurbitae* in the Okinawa islands of Japan [10]. Based on successful experiences of SIT as a control method for agriculture pests, it has also been evaluated for controlling the mosquito population in recent years, led by the Joint FAO/IAEA Centre of Nuclear Techniques in Food and Agriculture.

The incompatible insect technique (IIT) is also based on the same principles as SIT with the release of sterile insects. IIT relies on the strength of *Wolbachia*-induced cytoplasmic incompatibility (CI), which can be either unidirectional or bidirectional [11]. *Wolbachia pipiens* is a maternally inherited endosymbiont that is commonly found in insects, including several mosquito vector species [12,13]. IIT refers to the release of *Wolbachia*-infected male mosquitoes for copulation with wild-type females that contain no *Wolbachia* or a different strain of *Wolbachia*. As a result, the females will produce no offspring owing to CI [14]. Successful artificial transfer of *Wolbachia* between mosquito species makes IIT for mosquito control become a reality [15,16]. Field trials based on IIT have been performed to control *Ae**. albopictus* and *Ae**. aegypti* population numbers in the United States of America and Italy [17,18,19]. However, due to the imperfect sex separation of mosquitoes, inadvertent release of female mosquitoes carrying an artificial *Wolbachia* strain might lead to the unintentional establishment of *Wolbachia* in the field (population replacement), especially when the target population has been extremely suppressed. Interestingly, the combination of SIT–IIT can eliminate the population replacement risk caused by IIT, as the use of low-dose radiation can completely sterilize the females to prevent the spread of *Wolbachia* [20]. In addition, the applied dosage does not affect the mating performance of the sterilized male mosquitoes and their sterility strength when mating with wild females at the chosen release ratios [20,21,22]. The SIT–IIT strategy has been tested in open-field trials against *Ae. albopictus* in China [23] and *Ae. aegypti* in Thailand [24], with obvious suppression results that have been achieved without leading to population replacement. Another control strategy is based on the production of GMO mosquitos. It is generally referred to as antibiotic-based release of insects carrying a dominant lethal (RIDL) and has been used successfully in the open-field trials of *Ae. aegypti* control [25,26,27]. Though these methods follow different principles and may use different mosquito strains, the steps needed in SIT will also be applicable to these other methods.

The mosquito SIT involves several steps, including mass rearing, sex separation, irradiation, packing, transporting, releasing and monitoring [28]. The general process from package to sterile male release is shown in Figure 1. For packing, transport and release, there is a critical requirement: male mosquitoes must be chilled, held for a time and transported to the field without compromising the male’s quality, especially when SIT is applied to control mosquitoes in large-scale programs. In SIT programs against agricultural pests, sterile males are usually packed and transported in the chilling status, as the low temperature prevents them from moving. This not only facilitates the handling and packaging procedures but also reduces the physical damage to compacted insects [29,30]. Therefore, chilling and compaction conditions are required to be optimized for each mosquito strain to avoid affecting the quality of sterile males [31]. To accelerate the application of SIT for mosquito control on an operational scale, the Joint FAO/IAEA has promoted a coordinated research project (CRP) entitled “Mosquito Handling, Transport, Release and Male Trapping Methods” to investigate how to handle and transport sterile male mosquitoes with minimal influence on their quality, as well as develop highly efficient release, trapping and surveillance methods [32]. In this review, we summarize the current accumulated knowledge on mosquito handling, transport and release procedures, which are vital to the use of SIT and its related technologies against the major vector mosquito species throughout the world.

## 2. Pre-Release Procedures: Handling, Packing and Shipping Conditions

Cold tolerance is important for insect survival and dispersal. Rapid cold hardening (RCH) is caused by brief exposure to low temperatures [33], and it helps insects to survive when they are suddenly exposed to a low temperature. RCH has been found in numerous insects, including mosquitoes [34]. Due to RCH, a very common handling option for sterile males is chilling; some insects, such as fruit flies [35], can be chilled before release to facilitate handling for packing, transportation and release.

To facilitate the insects handling in the mass rearing facilities, insects are exposed to cold temperatures to be immobilized and packaged in special containers. As survival, flight ability and male mating competitiveness are the key parameters for evaluations of sterile males, it is essential to determine the tolerable and optimal range of chilling conditions, such as temperature, duration and male compaction rate under laboratory conditions [31,36]. A range of temperatures should be determined initially, with the lowest temperature not damaging adult quality and the highest value ensuring complete immobilization. For example, Culbert and colleagues found that when the temperature is above 12 °C, *An**. arabiensis* males are not anaesthetized [36]. Damages to mosquitoes are usually observed after exposure to low temperatures, and the injury is duration dependent. For instance, reduced survival is observed in *Ae. albopictus* males chilled at 1 °C for over 1 h, and the reduction is positively correlated with chilling duration [31]. A similar result is observed on *An. arabiensis* males chilled at 2 °C for over 24 h [36]. If the temperature range has been determined, a slight change of temperature values can be performed for further assessment. Due to the possibility of temperature fluctuations during transportation, the criterion for deciding a temperature is generally one to three degrees below the determined temperature range [31]. In addition, temperature is one of the factors that influence the recovery time for each mosquito species. When the temperature is low, longer recovery times will be required [31,36]. The longer the recovery period, the worse it is for the immobilized male mosquitoes because males lying on the ground are more vulnerable to predation and other forms of death [31]. After the chilling temperature is determined, the next step is to evaluate the maximum chilling duration for the male mosquitoes without greatly affecting the mating performance [31,36,37]. This is a key factor for area-wide strategies, as the area of coverage of a mosquito factory will depend on the maximum transport range, which in turn depends on the maximum duration of chilling.

When male mosquitoes are anaesthetized, chilling is frequently associated with packing methods [31,36,38]. Generally, males are placed in a container, and to quantify this procedure, the compaction rate is introduced, and it is regulated by altering the container’s interior space. The height of compaction will affect the survival of the male mosquitoes. For example, at the same storage density of 174.7 males/cm^3^, a lower survival of *Ae. albopictus* male adults was observed after compaction at a height of 8 cm compared to compaction heights at 2, 4 and 6 cm, respectively [31]. In the Ref [31] trial, the post-compacted male mosquitoes that could fly out of the storage containers were considered alive, while those that could not were considered dead. In the case of mosquitoes transported from New Mexico to California (USA), the males were maintained at 4 °C for about 20 h. It was found that more than 95% of the recovered male mosquitoes had lost their scales and had damaged wing edges after transport, but these damages did not affect the flight behavior. Interestingly, the results also showed that the higher compaction of male mosquitoes (240 males/cm^3^) exhibited higher survival after transport when compared to males maintained at lower compaction of 10 and 40 males/cm^3^. The potential explanation for this phenomenon was that the high compaction prevented male mosquitoes from colliding back and forth during transport and thus resulted in less damage [39]. The impact on the survival of *Ae. albopictus* male mosquitoes that were maintained at 8–12 °C for 1–1.2 h, with a compaction density of 69.4 males/cm^3^ and a height of less than 1 cm, has been investigated by Zhang and colleagues, and the results showed that no impact on survival was observed [31].

Though important, chilling and compaction are not necessary options in handling, as Crawford and colleagues showed in Ref [40], where an automated male mosquito release system that does not require chilling or compaction has been developed. As their trial demonstrated a higher efficacy than previous similar trials, this system may offer us an inspiring new way of sterile male handling.

Additionally, male quality control should be monitored on post-handled males after chilling or compaction. The mating competitiveness of sterile males is one of the key factors for the success of SIT. To induce sterility in the target wild population, sterile male insects have to compete with wild males to mate with as many wild females as possible. Thus, it is critical for sterile insects to have a good performance in the field. Chilling might have a negative impact on the mating competitiveness of males if the temperature and duration are not well optimized. Zhang and colleagues found that the mating competitiveness of *Ae. albopictus* males was reduced by about 50% when chilled at 10 °C for 24 h, while reduction was not observed in males chilled at the same temperature for only 3 h. Further, the authors also measured the glucose level of post-chilled males, and they found that decreased mating competitiveness of males, which were chilled at 10 °C for 24 h, may relate to the glucose reduction [31]. This method may solve the problem of the classical method being time consuming and laborious, and it can be quick in evaluating the mating performance of the post-chilled males. Another approach is to assess the flight ability of the sterilized males. The flight ability of insects is known to be a direct and reliable marker for evaluating insect quality, including mating. For example, the flight ability test has been demonstrated to be a good indicator of mating competitiveness in the sterilized fruit flies, tsetse flies and moths [30,35]. The flight cylinders, normally composed of PVC tubes, are used to estimate the flight ability. A new flight test device (FTD) has been designed by Balestrino and colleagues to estimate the survival and mating capacity of the radio-sterilized *Ae. albopictus* males by observing the escape rate of newly emerged adults from individual pupae [41]. To improve the practicality and response time of the FTD, Culbert and colleagues have developed a new flight cylinder device capable of testing 100 adults directly within 2 h [42]. Additionally, Ariane Dor and colleagues designed four devices to test males’ flight ability, and the result revealed the differences in flight ability of males of different ages and presented a recommended condition for flight ability test [43]. As the measurement is performed directly on adult males, the impact of various treatments, including mass rearing, irradiation, chilling, transportation and release, can be estimated apace and timely [44]. This newly developed tool could be a useful quality control method to evaluate cumulative stress through the whole handling process of males in SIT projects.

## 3. Transportation of Sterilized Male Mosquitoes

Usually, a rearing facility that produces sterilized insects may not be around the release areas; the facility and the release areas could be kilometers away. Therefore, it comes with the problem of transporting sterile males. Until now, mosquito transport in vehicles remains the main method in SIT projects. Male mosquitoes can be transported in a sober or anesthetized status, and the latter status can greatly reduce the required space for the sterilized male adults, thereby improving transport efficiency and reducing cost. Zhang and colleagues have compared three different ways of transporting *Ae. albopcitus* males, and the results indicated that at least 190 times less space was required for the anesthetized males than for sober males during transport [31].

Transport can be performed in the adult or pupa stage (Figure 1). For pupal transportation, it is generally more convenient to be transported at a certain low temperature, preventing the adult emergence during transportation [45]. It has been demonstrated that male pupae of *Ae. albopictus* can be stored at 15–18 °C for 20 h without eclosion [45], giving researchers enough time to transport them from the facility to the release site. In addition, stored male pupae normally emerge when transferred to room temperature within the next 48 h, and no impacts on the lifespan of the male adults were observed [45], suggesting that it is feasible for male pupae to be transported at a long distance under 15–18 °C. Other temperature conditions needed for transport have been discussed in the preceding chilling methods. Compared to adult mosquitoes, pupae are more resistant to stress, and therefore, their quality is less affected after handling or transport. However, the transport of mosquito pupae may face some abrupt occasions, such as the emergence ahead of time. If the rearing facility or the emergence center is far from the release site, a transfer center may be needed halfway, and the establishment of an emergence room at the transfer center may be an alternative option to solve the problem of early emergence (Figure 1). Setting up a cold room at the transfer center can also help chill the newly emerged male adults prior to transport and release. Additionally, the research conducted by Sasmita showed that the non-chilled conditions for transportation can be taken into consideration in a SIT operation [46]. Additionally, for transporting in the adult stage, the usual choice is to use vehicles when the mosquitoes are anaesthetized and at a certain compaction density, and they will always be given a 10% sucrose solution. Sometimes, the transportation is combined with the release procedure, thus the transportation conditions will be various, and the specific situation will be introduced afterward.

Additionally, the quality control (QC) tests for sterilized males delivered by shipment should be paid attention to. As Mastronikolos and colleagues showed, it is hard to transport mosquitoes over long distances, especially in cases where the duration is longer than 24 h. These authors tested several parameters of sterile males, including survival, flight ability, mating performance and competitiveness, and the results revealed negative impacts of a long transportation period on sterile males, such as the increased mortality; thus, it is critical to plan ahead of time for the efficient and quick transportation of mosquitoes [47].

Open-field trials have been reported for suppressing the *Aedes* mosquito population by the SIT-related strategies, summarized in Table 1, which shows the specific methods taken in each trial.

## 4. Release of Sterilized Male Mosquitoes

Before the release of the sterile male mosquitoes, it is required to obtain the permit from the local government and the support from the residents. The field trial experiments involved below are all licensed by the local government. In addition, release agreement is achieved from the residents in the release sites through community engagement.

### 4.1. Pupal Release

The first field trial of SIT application for mosquito control by pupal release was conducted by Bellini and colleagues to suppress the *Ae. albopictus* population in urban areas of Italy from 2005 to 2009. In this trial, male pupae produced in the laboratory were initially transported for irradiation and then to the release sites, which took 2–3 h of driving. Irradiated male pupae (24–30 h after pupation) were maintained in a plastic container with some water, and then, the container was placed near green plants and shelter in the release area. After emergence, male adults can fly out of the container to seek and mate with wild females. The release was performed once a week for 20–21 weeks, and about 2.1 million male pupae were released during the intervention in the point area, and the results revealed that releasing sterile males caused a considerable increase in sterility in the local population [32].

In the trial conducted in the Cayman Islands, for pupal release, male pupae were placed in release devices (such as paper cups) and allowed to emerge. Then, these devices were transported to the release sites three times a week, and the devices were open to allow males to disperse; lastly, dead pupae and adults were counted after retrieval. This trial undertook an extensive community engagement before conducting the biological experiments.

Pupae release has the following benefits: (1) it can reduce the labor of emergence in the mosquito factory; (2) it can reduce the handling process of adult mosquitoes, such as chilling and compaction, and therefore, potential damages; (3) it can reduce release frequency, as the process of emergence permits male mosquitoes to emerge continuously; (4) it is easier for most people to accept the release and not be bothered or irritated due to the use of adult males. However, there are several problems with pupae release too, such as: (1) newly emerged male mosquitoes cannot be provided with sugar solution in time, which may reduce the quality [54]; (2) newly emerged male mosquitoes will be more willing to stay around the appointed sites rather than fly away, which may result in poor dispersal in the release area compared to adult release; (3) pupae are sensitive to external interference, which makes them more liable to be damaged; (5) retrieval of the container of male pupae will be a massive undertaking, especially in a large-scale release. Therefore, adult release is more common in mosquito control projects.

### 4.2. Adult Release by Human Walking

Until now, the largest *Ae. albopictus* population control project using SIT–IIT was conducted in Guangzhou (China) under the leadership of Prof. Xi and colleagues from 2015 to 2017 [23]. In this project, sterile male mosquitoes produced in the facility were maintained in a release bucket (17 cm high × 17 cm diameter) with a density of 0.21–0.26 males/cm^3^ and then transported to the release sites in a vehicle for about 1 h. Sterile males were placed in a transfer center overnight and provided with 10% sucrose solutions. The release took place between 7:00 to 10:00 am the next day. The amount of sterile male mosquitoes released at each point is determined based on the suppression efficiency and the 5:1 release ratio. While the staff were walking through all the release sites, the container lids were open to allow the male mosquitoes to fly out freely. From 2016 to 2017, between 1.5 and 2.6 million male mosquitoes were released weekly on one site, while the number of males released weekly on another site was between 0.6 and 0.89 million, with a frequency of three times per week [23]. In this trial, non-irradiated HC males exhibited similar mating competitiveness to wild-type males, but when the mosquitoes were irradiated, they only had a competitive mating ability of 0.5 to 0.7 compared to wild-type males. Additionally, the authors mentioned that due to the reduced mating competitiveness, an increased number of mosquitoes released will be needed. In 2018, Xi’s team started to release the chilled male mosquitoes to maintain the suppression efficiency in both treated sites. The results indicated that the chilled males were as good as the non-chilled ones, as the mosquitoes would wake up from chilling during release via walking.

Another example of adult release for mosquito control was conducted in Chachoengsao Province, Thailand, where local researchers used the SIT/IIT to suppress the *Ae. aegypti* population [24]. Containers of 100 sterile male pupae were prepared and transported to the transfer center. After emergence, the adult mosquitoes were subsequently transported to the release site. Males were released by volunteers from the health department inside and outside each house in the treatment area at a frequency of once a week. In 2016, 10,000–25,000 sterile male mosquitoes were released weekly for a total of 24 weeks. The sterile males spread at an average distance of 379 ± 254.36 m and as far as 625 m in this experiment. Furthermore, the males lived an average of 5.80 ± 7.02 days, with some of them living up to 17 days in the field.

Moreover, in these two release trials, the authors both performed community engagement; meetings and seminars were held to introduce the principle and the process of the trial to the local governmental authorities and the residents. Obtaining governments’ permission and having support of the residents are quite vital to conducting a field trial in certain areas.

A similar release method was performed in Greece (SIT for *Ae. albopictus* control), Brazil (RIDL for *Ae. aegypti* control) and Mexico (IIT–SIT for *Ae. aegypti* control) in a mosquito control project [27,50,52].

### 4.3. Adult Release by Vehicles

A field trial against *Ae. aeypti* using IIT has been conducted in Fresno County, California (USA) [40]. In this project, researchers developed an automated mosquito release system to ensure the calibration and completeness of male mosquito distribution in the field areas. The automated system consisted of four parts: transport and release tubes (6-inch diameter), automated release devices attached inside customized vans, map-based release plan generation and triggering software, a structured light mosquito counter and an automatic packing system that does not require chilling or compaction [40], as stated before. The males were released in the field sites using two vans equipped with these automated release devices that can blow males through release outlets. The release was performed daily from 6:00 to 11:00 am for 26 weeks. The size of the treated area was 293 ha, and a total number of 14.4 million males released covered three replicate neighborhoods in the peak mosquito breeding season. As a result, the number of female mosquitoes was 95.5% lower in the release areas as compared to the control sites. The automated device made the mosquito release very precise and convenient and has not been proved to cause much damage to the males [40].

The release methods mentioned above are mainly conducted on the ground. It is clear that the existing ground release methods can be further improved [42], such as the design and capacity of the release units and the release mechanism. Additionally, release methods via public transport can be explored. However, the ground release may have some shortcomings. There are some inherent drawbacks to releasing sterile mosquitoes by vehicle or on foot, for example, the risk of contracting diseases by the operators in dengue-endemic areas and the cost of transport. The time and expenses required make the system difficult to upscale to routine-based releases of millions of mosquitoes.

### 4.4. Adult Release by Drone

Some agricultural insects, such as screwworm flies or Mediterranean fruit flies, are routinely released aerially. In fact, this method can be used for adult mosquitoes, and the disadvantages associated with ground release may largely be avoided by employing aerial release. The drones are quite an advantageous method of aerial release against vectors, since they can potentially fly over populated areas, while other vehicles, such as airplanes, are not usually allowed to, and the size of drones makes them flexible to perform the release. Considering the pursuit of large coverage areas, researchers take advantage of the emerging unmanned aerial vehicle (UAV) technology to release mosquitoes. Aerial release approaches will help ensure cost-effective releases of the sterile male mosquitoes with less staff required. In 2020, a fully automatic release system for the release of adult sterile male mosquitoes using an unmanned aerial vehicle, also known as a drone, was reported [55]. In this trial, the males were placed in a specialized cassette that was 5 cm in depth and carried no more than 50,000 males, as a density over 1.2 g/cm^3^ may cause injury. Additionally, the UAV was run at a speed of 10 m/s, while the cylinder’s speed was 2 rpm. Flight altitude was 50 m or 100 m, with the former showing better recapture rate, and the latter showing better dispersal during flight. Temperature was kept between 8 and 10 °C. This trial followed the Commission Implementing Regulation (EU) on the rules and procedures for the operation of unmanned aircraft and had the support of the national and local governments. The use of drones can result in improved area coverage and lower operational expenses due to the need for fewer release sites and less field labor. This new drone modality is expected to reduce the cost of SIT release studies by a factor of 20 [56]. For example, in the IIT–SIT trial against *Ae. albopictus* via ground release in China, the cost was estimated at USD 20 per hectare per week [23], while using a drone, it could be reduced to an estimated USD 1 per hectare per week [55]. Moreover, the recapture rate (0.32%) in Bouyer’s study was better than other trials whose releases were from the ground. As for the labor, compared with a drone, releases from the ground in the required sites in this study would have needed a vehicle, two field staff and 2 h work. The successful release of sterile males from a drone is quite an important outcome. This system can provide an optimized condition for the males to ensure their complete immobilization and compaction without doing harm, and it can also permit adequate flow release. On the other hand, this release system includes several components, such as an insulated storage unit, a release mechanism, a mechanism of ejection and onboard electronics, which can eject the mosquitoes onto the release area ramp and control the state of the mechanism and mosquitoes. The UAV can also be outfitted with a high-magnification camera, which can help visualize immature mosquitoes in small containers in the habitat. Moreover, the high-resolution drone mapping reported in 2021, which features vector larval habitats, can also help field release of mosquitoes. Such innovative methodology could be crucial for the integrated interventions of mosquito control [57,58]. Another trial in Mexico in which they used a drone to release sterile males has some differences in the design and function of the mosquito release device. They used a drone fitted with a tubular release container placed at a 45° angle on the bottom, and the release rate could be controlled electronically at ~80 males/second, while the drones were in “S”-shaped flight at a height of 50 m. The results showed better distribution, less time but lower capture numbers, indicating that the handling process needs some modifications to reduce injury. A series of meetings, assemblies and workshops were organized with the local authorities and residents before the trial began [59]. Though there are many advantages in using the UAV, there may be problems in the actual implementation, e.g., the regulatory status of the use of unmanned aerial vehicles over populated areas is uncertain in some countries. Before field trial, it is vital to carry out quality-control laboratory experiments of the release system to estimate its potential impact on the released sterile males. Additionally, MRR trials should be performed to define the proper release strategy, such as release pattern, altitude, flight speed, etc.

## 5. Future Perspectives

SIT mainly focuses on the hereditary machinery of insect populations rather than insecticides, habitat destruction or other pest control techniques, which offers a cost-effective and environmentally safe technology. With the existing knowledge of mosquitoes handling, transport and release, we may maximize the chances of success in SIT implementation. To improve transportation and release efficiency, there is no doubt that chilling and compaction are important for the handling of mosquitoes. Many studies have indicated that the qualities of male mosquitoes are not affected if the handling conditions are well optimized. However, the qualities of male mosquitoes are sharply reduced if the transportation time increases; ideally, the shipment time should be less than 24 h. The result emphasizes the requirement for a harmonized process regarding the global framework for the airfreight of sterile male mosquitoes, involving agreements among the involved countries. In addition, the release of sterile male mosquito program should be combined with geographic information system (GIS) and spatial analysis, as they can provide guidance on spatial sterile male mosquito requirements and dispersal patterns in relation to wild mosquito population densities, habitats, elevation, etc., in a dynamic way. Therefore, new kinds of vehicles and specialized devices can be designed to help with the implementation of SIT. However, from another point of view, one of the important constraints to adopting some technologies has been the high cost, so before the implementation of SIT, costs must be considered upfront as part of the decision-making process, depending on the context and specific program objectives. It can be ensured that with the industrialized process of mosquito facilities and the development of relevant machines or instruments, the implementation of SIT will be more precise and efficient. In the meantime, with several control tools or methods combined, a viable SIT program can be effective and achieve the best results. Moreover, optimizations should be conducted on mass rearing, sex separation and irradiation, as they are considered to play an important role in affecting the qualities of male mosquitoes, which are the important issues for the efficiency of SIT [60].

## 6. Conclusions

Existing vector control methods are not able to cope with the unprecedented emergence and re-emergence of vector-borne diseases. SIT and its related techniques under development have already shown dazzling effects in suppressing wild mosquito populations. The evidence for the effectiveness of these new technologies can be expected and may be used in larger field areas within the next few years. However, without optimized conditions and practical protocols for handling, transport and release of male mosquitoes, there is no way to make good use of these technologies. Laboratory results have indicated different conditions males need in handling and transport. Practical and convenient release methods have also been developed and evaluated in some pilot trials. Additionally, the successful release of sterile males by ground or by drone is an important outcome. With valid handling, transport and release techniques and local support, a widespread implementation may reverse the current alarming global disease trend.

## Figures and Tables

**Figure 1 insects-13-00532-f001:**
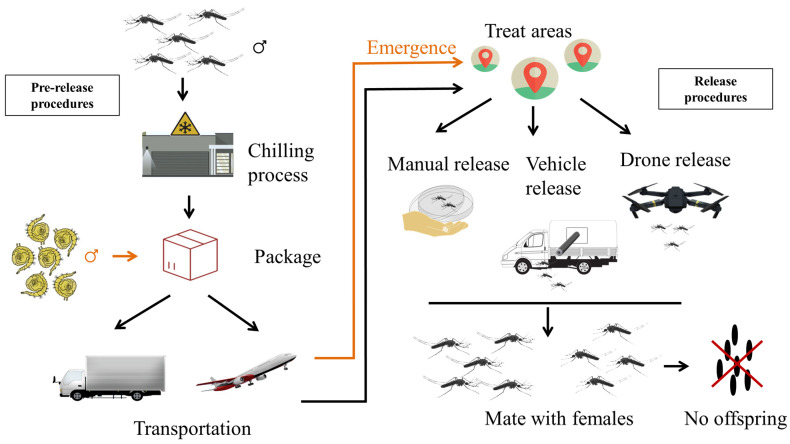
Handling procedures for shipping and releasing of sterile males in SIT mosquito projects. SIT is divided into two parts, the pre-release and the release procedures. The pre-release procedures include chilling, package and transportation. The release of sterile male mosquitoes can be performed either manually, by vehicle or drone. The black arrow shows the handling procedure of the adult mosquito, while the orange arrow adds the operations for the pupae.

**Table 1 insects-13-00532-t001:** Reported open-field trials of *Aedes* mosquito population control using SIT-related technologies.

Strategy	Target Mosquito Species	Country	Size of the Control Area (ha)	Release Duration (Months)	Transport Conditions	Release	Suppression Efficiency
SIT	*Ae. albopictus*	Italy	96 (total size in the five treated areas)	14	Pupae transport in plastic containers (12 cm diameter) by vehicles	Pupal release in plastic containers	Egg numbers, respectively, decreased 50.7%, 10.3%, 72.4% and 4.7% in four areas but increased 0.8% in one area [48]
	*Ae. albopictus*	Mauritius	3	8	3-day-old adult transport of 2000 males/cage (30 × 30 × 30 cm) covered with wet towels by vans at ambient temperature	Adult release	Female numbers decreased 28.6% (Min) to 88.2% (Max) [49]
	*Ae. albopictus*	Greece	5	2	Adult transport of 1000–1500 males/box by vehicles	Adult release by opening boxes while walking	Egg hatch rate decreased 40–84% without showing a decrease trend in egg numbers [50]
	*Ae. aegypti*	Cuba	50	5.5	Pupae transport of 6000 males/cardboard box (15 × 15 × 60 cm) by vehicles	Adult release by opening boxes while vehicles moving	Egg numbers significantly decreased, and no viable eggs were collected for up to 6 weeks [51]
IIT	*Ae. albopictus*	USA (Kentucky)	12.5	4.25	Adult transport in cardboard mailing tubes (about 5 cm)	Adult release	Egg hatch rate and female numbers significantly decreased [18]
	*Ae. albopictus*	USA (Miami)	68.8	5	<48 h adult transport of 1000 males/tube (about 5 × 30 cm) via commercial courier in a cooler with moistened towel and a temperature sensor	Adult release	Egg hatch rate decreased 32–62%, and female numbers decreased 78% (Max) [19]
	*Ae. albopictus*	Italy	2.7	1.5	1–2-day-old adult transport of 750 males/cage by cars	Adult release	Maximum 16% difference was observed in egg hatch rate [17]
	*Ae. aegypti*	USA (California)	293	6.5	Adult transport in release tubes (6-inch diameter) by cars	Adult release by automated release system	Female numbers decreased 95.5% [40]
	*Ae. aegypti*	Mexico	50	6	Adult transport at 22 °C in plastic cylinder vases (2.8 L) by a van	Adult release by a team	Suppression efficacy was 90.9% a month after initiation of the suppression phase, 47.7% two months after, 61.4% four months after, 88.4% five months after and 89.4% at six months [52]
SIT–IIT	*Ae. albopictus*	China	32.5	16–23	Adult transport of 1000 males/release bucket (17 cm diameter × 17 cm height) by vans	Adult release in release bucket	Egg numbers decreased more than 94%, and female numbers decreased 83–94% [23]
	*Ae. aegypti*	Thailand	65	6	Pupae transport of 100 males/container by vehicles	Adult release	Egg hatch rate decreased 84%, and female numbers decreased 97.3% [24]
RIDL	*Ae. aegypti*	Cayman Islands	103	5.75	Pupae transport in release devices by vehicles	Pupal release and adult release	Larval numbers decreased 80% [53]
	*Ae. aegypti*	Brazil	11	1.5	Adult transport of 500–1000 males/release device (14 cm high × 13 cm diameter) by truck	Adult release by opening release devices on vehicles	Female numbers decreased 95% [27]
	*Ae. aegypti*	Panama	10	1	Adults transport of 1000 males/pot (14 cm high × 13 cm diameter) in transport boxes by vehicles	Adult release by opening plastic container on vehicles	Female numbers of *Ae. aegypti* (target mosquito species) decreased 91–95% without affecting the abundance of *Ae. albopictus* [26]

## Data Availability

No new data were created or analyzed in this study. Data sharing is not applicable to this article.

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
