# Peer review of "Current Status of Mosquito Handling, Transporting and Releasing in Frame of the Sterile Insect Technique"

_insects, 2022, doi:10.3390/insects13060532_

Round 1

Reviewer 1 Report

MS_insects-1748262

Review report

The SIT strategy that relies on release of radiation sterilized males raises hopes in the fight against major vector-borne diseases and fast forward to the last decade its development continues at a sustained pace around the world. Developing any mosquito SIT program require a substantial R&D component which addresses several issues including developing of mass-rearing system, improving radiation procedures for mosquito sterilization, handling, optimal holding conditions to transport sterile mosquitoes over long distances without loss of quality and release methodology. The main purpose of this paper seems to be a review of recent advances on the last three technical aspects (i.e. handling, transport and release), which are critical to support a successful SIT application. The authors also share some insights into the latest successes obtained in Italy, Greece, China, Thailand, Cuba, Panama, Brazil, USA, etc.  The long list of cited references shows that there is ever-increasing accumulation of evidence that the SIT approach can be technically and logistically possible for vector control, and that when judiciously applied, it can be effective for the suppression Ae. albopictus and Ae. aegypti.

Guo and colleagues argue that each of the supportive pre-release techniques has its own specific advantages and limitations and that these (technical) limitations to SIT processes demand the research effort to develop appropriate quality assurance procedure. Indeed, the quality of released males remains a key concern, regardless of the specific handling, transportation and release processes, as processes such as chilling, compaction,  long distance transportation of large number of viable sterile  mosquitoes whether pupae or adults, etc  may severely compromise the quality and competitiveness of insect.

Although the application of tools and processing technologies to increase efficacy of SIT has technical and operational challenges, the authors conclude that improvement of these steps should offer potential to make a significant contribution to the vector suppression and ultimate prevention of emerging and re-emerging mosquito-borne diseases.

Overall, this article is informative in their current format, and should be published.  I only have a few minor points for authors to consider for improvement.

  • P2, first line: change “breeding animal” to ‘livestock’
  • P2, third paragraph, line 88: “In addition, the dosage does not affect….” Please write “ In addition the applied irradiation dose does not affect…”
  • P3, 1st paragraph, instead of “The general process from package to release of SIT is as Figure 1.”, please  write “The general process from package to sterile male release is as shown in Figure 1.
  • P.3 Figure 1. Please add a legend to briefly describe what is depicted.  What is meant by “Meet females”? should be ‘Mate with females”
  • P4. The sentence “Due to the possibility of temperature fluctuations during transportation, the criterion for picking a temperature is generally one to three degrees below the temperature range” is difficult to understand. Change “picking” to ‘deciding’. Do you mean below the AMBIANT temperature? 
  • P5. Transportation of sterilized male mosquitoes”, line 215 in the second paragraph: “For transporting in the pupa stage, it is generally to transport pupae at a certain low temperature…” write …”it is generally more convenient to… Perhaps you should also emphasize that the chilling (pupae and adults), would be necessary only for long distance transport from the production facility to field site, and that chilling and compaction of adults are important conditioning processes for aerial release, thereby improved the release efficiency.
  • Table 1.  Although it would be difficult to compare SIT practices among contexts with different capacities and resource availability, it would be worth including the possible operational limitation of each cited studies in table 1, in terms of processing, transportation after the irradiation and ground vs. aerial releases of pupae vs adult. A short discussion could be given to make it easier for the reader to understand how specific processes in all steps have influenced the outcome in each context.

·        The authors repeatedly mention that establishing the emerging technological options for handling, packaging, transporting and releasing sterile males and monitoring “is expected to reduce the cost of SIT” process, besides having a strong potential to influence the effectiveness of intervention.  However, the authors could also emphasize that a significant constraint to the adoption of some of the sophisticated technologies (e.g. mass-rearing infrastructure and operation, irradiation, transportation, aerial release) has been their generally higher cost. A logical recommendation the authors could make (e.g. in p11, section 5 on Future perspectives) is that costs of establishing such capacity must be considered up-front as part of the decision-making process, depending on the context and specific program objectives.

The authors should stressed (for ex. at the end of section 5) that, although it is important to establish appropriate technical capacity and logistical feasibilities in most circumstances, viable SIT program can be effective only when several (compatible) control tools or methods are combined.

Author Response

We thank Reviewer 1 for his positive comments on our submitted review. Below is our responses to Reviewer 1's questions. 

Reviewer #1: P2, first line: change “breeding animal” to ‘livestock’

Response: We have revised the word, see line 47.

P2, third paragraph, line 88: “In addition, the dosage does not affect….” Please write “ In addition the applied irradiation dose does not affect…”

Response: We have add the word, see line 85.

P3, 1st paragraph, instead of “The general process from package to release of SIT is as Figure 1.”, please  write “The general process from package to sterile male release is as shown in Figure 1.’

Response: We have revised the sentence, see lines 95-96.

P.3 Figure 1. Please add a legend to briefly describe what is depicted.  What is meant by “Meet females”? should be ‘Mate with females”

Response: We have added some descriptions under Figure 1 (see lines 114-117) and revised the “Meet females” to “Mate with females”.

P4. The sentence “Due to the possibility of temperature fluctuations during transportation, the criterion for picking a temperature is generally one to three degrees below the temperature range” is difficult to understand. Change “picking” to ‘deciding’. Do you mean below the AMBIANT temperature?

Response: Here we want to express that because of the temperature may fluctuate, so the chosen temperature can be lower to handle this situation. We have revised the sentence to “Due to the possibility of temperature fluctuations during transportation, the criterion for deciding a temperature is generally one to three degrees below the determined temperature range”. See lines 137-139.

P5. “Transportation of sterilized male mosquitoes”, line 215 in the second paragraph: “For transporting in the pupa stage, it is generally to transport pupae at a certain low temperature…” write …”it is generally more convenient to…

Response: We have revised the sentence. See lines 209-210.

Perhaps you should also emphasize that the chilling (pupae and adults), would be necessary only for long distance transport from the production facility to field site, and that chilling and compaction of adults are important conditioning processes for aerial release, thereby improved the release efficiency.

Response: We thank R1’s for the valuable comments. We believe that chilling is required not only for long distance transportation, but also for short distance as mosquitoes can be compacted under chilling conditions, which will improve the transportation efficiency (more male can be shipped per time). In lines 215-216, we have stated that pupal transportation under chilling conditions can be performed for long distance “suggesting that it is feasible for male pupae to be transported for a long distance under 15-18 ℃”.

Table 1.  Although it would be difficult to compare SIT practices among contexts with different capacities and resource availability, it would be worth including the possible operational limitation of each cited studies in table 1, in terms of processing, transportation after the irradiation and ground vs. aerial releases of pupae vs adult. A short discussion could be given to make it easier for the reader to understand how specific processes in all steps have influenced the outcome in each context.  

Response: We agree with R1’s comments that it is better to list the limitations of each reported field trials. However, it is quite difficult to compare for each trial as the information such as the mass rearing efficiency, cost, treated size, distance from the facility and treated areas, human resources, ect, is provided insufficient in the articles. Thus we only list the suppression efficiency achieved in Table 1 in each field trial as we believe this is the most important parameter for comparison.

The authors repeatedly mention that establishing the emerging technological options for handling, packaging, transporting and releasing sterile males and monitoring “is expected to reduce the cost of SIT” process, besides having a strong potential to influence the effectiveness of intervention. However, the authors could also emphasize that a significant constraint to the adoption of some of the sophisticated technologies (e.g. mass-rearing infrastructure and operation, irradiation, transportation, aerial release) has been their generally higher cost. A logical recommendation the authors could make (e.g. in p11, section 5 on Future perspectives) is that costs of establishing such capacity must be considered up-front as part of the decision-making process, depending on the context and specific program objectives.

Response: We thank R1’s for the valuable comments. We have revised the paragraph and added the proposed opinion, see lines 398-401.

The authors should stressed (for ex. at the end of section 5) that, although it is important to establish appropriate technical capacity and logistical feasibilities in most circumstances, viable SIT program can be effective only when several (compatible) control tools or methods are combined.

Response:  We thank R1’s for the valuable comments. We have revised the paragraph and added the proposed opinion, see lines 403-405.

Reviewer 2 Report

Overall it´s a well-written review that addresses advances in the handling, transport, and release of males in programs or strategies that use mass rearing of mosquitoes, such as the sterile insect technique or the incompatible insect technique. I suggest some minor comments to improve the understanding of this review.

Simple Summary

No comments.

Abstract

No comments.

Introduction

No comments.

Pre-release procedures: handling, packing and shipping conditions

Lines 172-176: The wording is not very clear. It needs to be rewritten or rephrased.

Transportation of sterilized male mosquitoes

Lines 240-243: What were the negative impacts on transported sterile males?

Release of sterilized male mosquitoes

Pupal release

Lines 269-272: The release of male pupae is very discreet compared with the adult sterile male releases, which can engender the SIT rejection. So, another benefit may be that most people accept the use of the SIT and are not bothered or irritated as in the adult male releases.

Adult release by human walking

Lines 290-291: Unclear sentence

Lines 294-296: What were the mating competitiveness index values of non-irradiated males and wild-type males?  Non-irradiated males, were Wolbachia-infected male mosquitoes?

Lines 306-311: In a recently published study, radiation-sterilized Ae. aegypti males combined with Wolbachia (wAlbB) were released every 100 m at release points outdoor the home. The study was conducted for 6 months and achieved high levels of suppression. "Pilot trial using mass field-releases of sterile males produced with the incompatible and sterile insect techniques as part of integrated Aedes aegypti control in Mexico". PLoS Negl Trop Dis

Lines 311-315: Here and elsewhere in "Release of sterilized male mosquitoes." It suggests that the authors make a subsection in which the topics related to permits with authorities and residents, community engagement, levels of acceptance of the SIT, etc., are addressed.

Adult release by vehicles

No comments.

Adult release by drone

Lines 352-358: There is another recently published study in which they used a drone to release sterile males that have notable differences from the cited article in this review. The differences between these two studies are mostly in the design and function of the mosquito release device attached to the aerial drone, but also in the design of the flight path, duration of the field experiments, size of study areas, and the number of irradiated male mosquitoes released. "Comparison of ground release and drone-mediated aerial release of Aedes aegypti sterile males in Southern Mexico: Efficacy and challenges". Insects

Future perspectives

No comments.

Conclusions

Line 409: vector-borne diseases

Lines 411-412 Could it be used in area-wide SIT-based suppression programs?

Author Response

We thanks Reviewer 2 for his positive support on our submitted review. Below is our responses to the raised questions: 

Reviewer #2: Pre-release procedures: handling, packing and shipping conditions

Lines 172-176: The wording is not very clear. It needs to be rewritten or rephrased.

Response: We have revised the sentence to make it clear. See 181-182.

Transportation of sterilized male mosquitoes

Lines 240-243: What were the negative impacts on transported sterile males?

Response: We have added some information in line 237.

Pupal release 

Lines 269-272: The release of male pupae is very discreet compared with the adult sterile male releases, which can engender the SIT rejection. So, another benefit may be that most people accept the use of the SIT and are not bothered or irritated as in the adult male releases.

Response: We have revised the paragraph and added proposed opinion in lines 268-269.

Adult release by human walking

Lines 290-291: Unclear sentence

Response: We have revised the sentence, see lines 296. 

Lines 294-296: What were the mating competitiveness index values of non-irradiated males and wild-type males?  Non-irradiated males, were Wolbachia-infected male mosquitoes?

Response: We have revised the sentence, see lines 289-290.

Lines 306-311: In a recently published study, radiation-sterilized Ae. aegypti males combined with Wolbachia (wAlbB) were released every 100 m at release points outdoor the home. The study was conducted for 6 months and achieved high levels of suppression. "Pilot trial using mass field-releases of sterile males produced with the incompatible and sterile insect techniques as part of integrated Aedes aegypti control in Mexico". PLoS Negl Trop Dis

Response: We have revised the paragraph and updated the table 1 as well as the reference.

Lines 311-315: Here and elsewhere in "Release of sterilized male mosquitoes." It suggests that the authors make a subsection in which the topics related to permits with authorities and residents, community engagement, levels of acceptance of the SIT, etc., are addressed.

Response: We thank R1’s for the valuable comments. One paragraph has been added to express the permit and support before the release, see lines 245-248.

Adult release by drone

Lines 352-358: There is another recently published study in which they used a drone to release sterile males that have notable differences from the cited article in this review. The differences between these two studies are mostly in the design and function of the mosquito release device attached to the aerial drone, but also in the design of the flight path, duration of the field experiments, size of study areas, and the number of irradiated male mosquitoes released. "Comparison of ground release and drone-mediated aerial release of Aedes aegypti sterile males in Southern Mexico: Efficacy and challenges". Insects

Response: We have revised the paragraph and cited this article.

Line 409: vector-borne diseases

Response: We have revised the word, see line 408.

Lines 411-412 Could it be used in area-wide SIT-based suppression programs?

Response: We check our submitted ms and it is the reference 5 in lines 411-412. We think R2 might refer the wrong lines so that we cannot answer this question.

Reviewer 3 Report

Insects- 1748262

Current status of mosquito handling, transporting and releasing in frame of the sterile insect technique.

Comments for the Authors

The manuscript summarizes in a very general way the critical points of handling, transport and release conditions of sterilized male mosquitoes focusing on the SIT.

For a better understanding of this review, it would be very useful to briefly review the background on the effects on the quality of pre-irradiation and irradiation processes, or at least cite a manual or guide or articles on these aspects.

Once addressing this issue, I recommend to accept this manuscript with minor corrections.

Specific comments or suggestions.

L 117. Figure 1. Handling procedures for shipping and releasing of sterile males in SIT mosquito projects. It is important to differentiate the strategies that have been used (SIT, IIT, etc.).

L 247. Table 1. Reported open-field trials of Aedes mosquito population control using SIT-related technologies. It would be helpful to add the release costs and density of insects per hectare, as well as homogenize the Suppression efficiency column.

L 127. I will suggest to review and cite the following paper: Dor, A., Maggiani-Aguilera, A. M., Valle-Mora, J., Bond, J. G., Marina, C. F., & Liedo, P. (2020). Assessment of Aedes aegypti (Diptera: Culicidae) Males Flight Ability for SIT Application: Effect of Device Design, Duration of Test, and Male Age. Journal of Medical Entomology, 57(3), 824-829.

Author Response

Comments for the Authors

The manuscript summarizes in a very general way the critical points of handling, transport and release conditions of sterilized male mosquitoes focusing on the SIT.

For a better understanding of this review, it would be very useful to briefly review the background on the effects on the quality of pre-irradiation and irradiation processes, or at least cite a manual or guide or articles on these aspects.

Once addressing this issue, I recommend to accept this manuscript with minor corrections.

Response: We thank Reviewer 3 for the positive comments on our submitted review. We have added one sentence to strength the importance of pre-irradiation and irradiation processes and one new reference. See lines 405-407 and reference 60.

L 117. Figure 1. Handling procedures for shipping and releasing of sterile males in SIT mosquito projects. It is important to differentiate the strategies that have been used (SIT, IIT, etc.).

Response: We thank R3 for his comments. The difference between SIT and IIT is the sterilization method is different, using the irradiation or Wolbachia. However, the purpose of figure 1 is to show the pre-release (mainly on handling, chilling and transportation) and release procedures, not includes the mass rearing, sex separation and sterilization, so that we would like to keep the figure 1.

L 247. Table 1. Reported open-field trials of Aedes mosquito population control using SIT-related technologies. It would be helpful to add the release costs and density of insects per hectare, as well as homogenize the Suppression efficiency column.

Response: We agree with R3 for his comments on evaluating the cost of each field trial, however, not all references mention the costs used, so we do not add these information in the table. As we focus the release procedure, we have listed the cost on different release method, see lines 354-356.

L 127. I will suggest to review and cite the following paper: Dor, A., Maggiani-Aguilera, A. M., Valle-Mora, J., Bond, J. G., Marina, C. F., & Liedo, P. (2020). Assessment of Aedes aegypti (Diptera: Culicidae) Males Flight Ability for SIT Application: Effect of Device Design, Duration of Test, and Male Age. Journal of Medical Entomology, 57(3), 824-829.

Response: We thank R3 for his comments and we have cited the proposed article. See Reference 43.